# Nocellara Del Belice (*Olea europaea* L. Cultivar): Leaf Extract Concentrated in Phenolic Compounds and Its Anti-Inflammatory and Radical Scavenging Activity

**DOI:** 10.3390/plants12010027

**Published:** 2022-12-21

**Authors:** Vincenzo Musolino, Roberta Macrì, Antonio Cardamone, Maria Serra, Anna Rita Coppoletta, Luigi Tucci, Jessica Maiuolo, Carmine Lupia, Federica Scarano, Cristina Carresi, Saverio Nucera, Irene Bava, Mariangela Marrelli, Ernesto Palma, Micaela Gliozzi, Vincenzo Mollace

**Affiliations:** 1Laboratory of Pharmaceutical Biology, Department of Health Sciences, Institute of Research for Food Safety & Health IRC-FSH, University “Magna Græcia” of Catanzaro, 88100 Catanzaro, Italy; 2Department of Health Sciences, Institute of Research for Food Safety & Health IRC-FSH, University “Magna Græcia” of Catanzaro, 88100 Catanzaro, Italy; 3Department of Pharmacy, Health and Nutritional Sciences, University of Calabria, 87036 Rende, Italy

**Keywords:** oleaceae, *Olea europaea* L., waste by-product, leaf extract, polyphenolic compounds, oleuropein, luteolin-7-O-glucoside, electron paramagnetic resonance (EPR), non-alcoholic fatty liver disease/metabolic associated fatty liver disease (NAFLD/MAFLD), antioxidant anti-inflammatory activity

## Abstract

*Olea europaea* L. is a plant belonging to the Oleaceae family, widely grown around the Mediterranean Basin and its leaves are a source of phenolic compounds with antioxidant and anti-inflammatory capacity. Among these, oleuropein and luteolin-7-O-glucoside represent two major polyphenolic compounds in olive-leaf extract. Herein, a polystyrene resin was used to recover the polyphenolic fraction from the acetone-water leaf extract from Nocellara del Belice cultivar, which showed the higher level of analysed bioactive compounds, compared to Carolea cultivar. The antioxidant activity of the extract concentrated in phenolic compounds (OLECp) was evaluated through a classical assay and electron paramagnetic resonance (EPR) for DPPH and hydroxyl radicals scavenging. Thus, the anti-inflammatory activity and the potential beneficial effects in reducing lipid accumulation in an in vitro model of NAFLD using McA-RH7777 cells exposed to oleic acid (OA) were evaluated. Nile Red and Oil Red O have been used to stain the lipid accumulation, while the inflammatory status was assessed by Cytokines Bioplex Assay. OLECp (TPC: 92.93 ± 9.35 mg GAE/g, TFC: 728.12 ± 16.04 mg RE/g; 1 g of extract contains 315.250 mg of oleuropein and 17.44 mg of luteolin-7-O-glucoside) exerted a good radical scavenging capability (IC_50_: 2.30 ± 0.18 mg/mL) with a neutralizing power against DPPH and hydroxyl radicals, as confirmed by the decreased signal area of the EPR spectra. Moreover, OLECp at concentration of 25, 50 and 100 μg/mL counteracted the intracellular inflammatory status, as result of decreased intracellular lipid content. Our results highlighted the multiple properties and applications of an *O. europaea* extract concentrated in polyphenols, and the possibility to formulate novel nutraceuticals with antioxidant properties, destined to ameliorate human health.

## 1. Introduction

*Olea europaea* L. is an evergreen tree or shrub native to the Mediterranean basins, probably to Asia Minor [1,2]. The Mediterranean form, *O. europaea*, subspecies *europaea*, includes wild (*O. europaea* subsp. * europaea* var. *sylvestris*) and cultivated olives (*O. europaea* subsp. *europaea* var. *europaea*). The olive genus *Olea* includes 40 taxa within the family Oleaceae, distributed in three subgenera (*Paniculate, Tetrapilus* and *Olea*) [3,4]. Nowadays, there are more than 2000 cultivars in the Mediterranean basin, which are the result of a long process of domestication and selection conducted to obtain fruits with distinct morphological characteristics and plants capable of adapting to different climate conditions. Indeed, *O. europaea* L. is one of the widely cultivated fruit crops in the world, for the production of olive oil and table olives [5]. In Italy, Calabria represents the second Italian region for olive production and has unique varieties of cultivars. Among 33 native cultivars, the most common are Carolea, Tondina, Roggianella, Cassanese, Moresca, Grossa di Gerace, Ottobratica, Dolce di Rossano and Sinopolese [6,7]. Furthermore, valuable allochthonous varieties, such as Coratina, Leccino, Frantoio and Nocellara del Belice, are widespread in this region [7].

*O. europaea* L. ripe or near-ripening fruits contain water, protein, carbohydrate, cellulose, inorganic substances, phenolic compounds, pectin, organic acids, pigments and up to 22% oil [8]. In the drupes, the oil is mainly concentrated at the level of the fleshy mesocarp. The oil is a lipophilic phytocomplex obtained pressing the paste deriving by grinding the fruits or separating the oil from the pulp by centrifugation. The oil is mostly characterized by two components, the saponifiable fraction, representing 98.0–98.5% of the entire chemical composition, composed by triacylglycerols, partial glycerides, esters of fatty acids or free fatty acids, phosphatides, waxes and an unsaponifiable fraction consisting of tocopherols, phytosterols, pigments and phenolic compounds, representing around 1–2% of the oil composition [9]. Because of the large amount of monounsaturated fatty acid, such as oleic acid, the consumption of olive-derived products is highly recommended for a healthy lifestyle and a milestone in the Mediterranean diet [10]. Currently, the process of olive oil production generates a large amount of waste, including mill wastewaters, pulp and olive’s leaves. Leaves from *O. europaea* are treated as a waste from the trees, because of processing and pruning practices and, annually, a large amount of them is produced. In particular, the estimation of *O. europaea* L. leaves amount, collected in the main olive oil producers’ countries, could be around 750,000 tons per year [11]. Due to the high numbers, this waste by-product should be reused to promote a circular economy, as an example, to produce bioenergy and biofuels, for agricultural uses and even more, for the extraction of precocious bioactive compounds [12]. 

Based on molecular profile, the bioactive compounds extracted from olive leaves are grouped in acids, lignans, phenols, flavonoids and secoiridoids [13]. Among the secoiridoids, oleuropein represents the major compound of *O. europaea* L. leaves, which can constitute up to 6–9% of dry matter in the leaves, and the main compound responsible for the antioxidant and bioactive properties of hydroalcoholic extracts [14]. Furthermore, the compositional analysis of olive leaf extract also underlined luteolin-7-O-glucoside as another major polyphenolic in olive leaves [15]. It has been demonstrated that luteolin-7-O-glucoside could have gastroprotective effects in animal models of gastric ulcers [16]. Certainly, *O. europaea* leaf extract could have a pleiotropic effect mainly attributed either to the total phenols or to the individual phenolic compounds and to flavonoids and their derivatives, that have powerful antioxidant and anti-inflammatory activities in culture cells [17], invertebrates [18] and mammals [19,20]. In traditional herbal medicine, *O. europaea* leaves have been extensively used to reduce blood sugar, cholesterol, uric acid, diarrhoea, respiratory and urinary tract infections and gastro-intestinal diseases [21]. These empirical data are confirmed and extended through recent studies that highlighted the very important beneficial effects concerning the prevention of the main cardiometabolic risk factors (i.e., dyslipidaemia and insulin resistance) [22,23]. In this scenario, the prevention of non-alcoholic fatty liver disease (NAFLD) development represents a key factor to limit metabolic syndrome complications [24,25,26].

Among Calabrian olive cultivars, leaf-derived bioactive molecules could differ in quality and quantity, not only in relation with the properties of olive products, but also with the presence of metabolites exerting the biological activities [27,28]. Thus, the choice of the most appropriate extraction method, to maximize their yield optimizing its cost, represents a key factor to their use in industrial processes. To extract phenolic compounds from olive leaves, conventional extraction methods, using organic solvents, have been traditionally used [29]. Maceration is a solid–liquid extraction technique that involves extraction, by keeping the sample immersed in the solvent, to render soluble the bioactive compounds contained in the samples. Although maceration is a cheaper method compared to new developed techniques, it has some limitations, such as the timing of extraction [30]. Besides the extraction method, the type of the solvent plays a crucial role in the recovery of phenolic compounds; indeed, among organic solvents, ethanol, acetone and their water mixtures have been shown to be the best choice for the extraction of polyphenols [31]. Furthermore, it is well known that the concentration of the used solvent could affect the extraction yields, highlighting an higher recovery of polyphenols as well as a major antioxidant power of the extract when 50% acetone or 50% ethanol are used [32,33]. However, despite the advantages obtained using the organic solvent, they may not be efficient; thus, the employment of different assistive novel extraction technologies (i.e., sonication, microwave, pressurized liquid extraction or superfluid liquid extraction) could be very effective and time and cost-saving [32,34,35,36,37,38]. Ultrasound sonication is a technique in which samples are mixed with organic solvents in flasks; during the process, the produced sound waves lead to the release of polyphenolic compounds by rupturing the cell walls of the leaves. It has been reported that ultrasound-assisted extraction is faster and more efficient than other traditional extraction methods, such as maceration. Indeed, normally, about twenty min are required for ultrasound-assisted extraction versus more than 12-h for the maceration extraction method [39]. Based on this evidence and to enhance the efficiency of extraction, a 3-h maceration protocol, using acetone/water or ethanol/water, followed by sonication, to obtain a higher total phenolic content [33], has been applied. 

The aim of our work was to compare the phytochemical profile of two leaf extracts from Nocellara del Belice and Carolea cultivars through: (a) an efficient as well as less expensive extraction; (b) a method to concentrate extracted polyphenols to increase the yield in bioactive compounds.

Furthermore, to assess their biological properties, the potential beneficial effects in reducing oxidative stress, inflammation and lipid accumulation in an in vitro model of NAFLD using McA-RH7777 cells exposed to oleic acid (OA) were investigated. After the comparison, the radical scavenging power of Nocellara del Belice cultivar was assessed using Electron Paramagnetic Resonance (EPR) that represents the unique technique able to directly detect, identify and measure radical species. As the hydroxyl radical production has a fundamental role in the pathogenesis of the cited mechanisms underlying NAFLD, for the first time, the scavenging activity of hydroxyl radical has been specifically studied.

## 2. Materials and Methods

### 2.1. Plant Material

*O. europaea* L. leaves, from Nocellara and Carolea cultivars (Figure 1), were harvested in June 2021. All the species were collected in the geographical area of Catanzaro (Calabria, Italy). The taxonomic identification was confirmed by Dr. C. Lupia, Department of Health Sciences, University “Magna Graecia” of Catanzaro. Voucher specimens were deposited in the Mediterranean Ethnobotanical Conservatory (Sersale, Catanzaro, Italy) under the following accession numbers *O. europaea* L. cv. Nocellara del Belice: 17; *O. europaea* L. cv. Carolea: 18.

### 2.2. Extraction Procedure

*O. europaea* leaves from the two cultivars were processed separately. After the collection, they were placed in the dark, dried at +40 °C for 72 h and then ground in liquid nitrogen with a mortar and pestle. The minced leaves were transferred in a 200 mL glass flask and macerated at room temperature (RT), in the dark, for 3 h, using mixtures of acetone/water (ace/w, 50/50, % *v*/*v*) or ethanol/water (EtOH/w, 50/50, % *v*/*v*). Leaves/solvent ratio was 1 g/20 mL Following maceration, each solution was sonicated for 20 min in ice. The sonication was conducted using an ultrasonic homogenizer (Sonopuls model HD 2070, Bandelin, Berlin, Germany) equipped with a titanium-alloy flat-tip probe (13 mm in diameter; VS 70 T, Bandelin, Berlin, Germany) with a frequency of 20 kHz. The amplitude was controlled to set the ultrasonic output power at 70% of the maximum range. Continuous modes of operation were used for extraction (10 cycles corresponded with the continuous mode of operation) [40]. During ultrasonic treatment, to prevent the reaching of high temperatures and the subsequent degradation of phenolic compounds, we carried out the partial submerging of glass flask into a cooling bath, filled with ice. Then, each solution was centrifuged at +4 °C, 1036 rcf for 5 min and subsequently filtered through a filter paper. The extract was evaporated under vacuum using a rotary evaporator, to obtain a dry crude extract.

### 2.3. HPLC Analysis

Among the active constituents, the identification of oleuropein and luteolin-7-O-glucoside was performed by HPLC. HPLC analysis was performed using a PerkinElmer Flexar Module equipped with a photodiode-array (PDA) detector, a series 200 autosampler, a series 200 peltier LC column oven, a series 200 LC pump and an Agilent 4 μ C18 100A (250 × 4.6 mm) column. HPLC analysis and the data collection was carried out online using Chromera software (version 3.4.0.5712, PerkinElmer Inc., Waltham, MA, USA). An amount of 10 mg of dry extract was dissolved in 10.0 mLof a mixture of ace/w 1:1 *v*/*v*. The solution obtained was vortexed till complete dissolution, then filtered with a 0.2 μm PTFE filter. Finally, 10 μL of sample was injected into the HPLC system. A two solvent gradient (0.88% trifluoroacetic acid/acetonitrile) was used for the elution with a flow of 1.1 mL/min. The column temperature was set at 30 °C. The detector wavelength was set at 280 nm. HPLC analysis was performed before the enrichment of the extract of bioactive compounds, to choose the best extract in term of oleuropein and luteolin-7-O-glucoside, and after the loading on polystyrene resin column.

### 2.4. Preparation of O. europaea Leaf Extract Concentrated in Polyphenols

*O. europaea* leaf ace/w extract, containing the highest amount of oleuropein and luteolin-7-O-glucoside (Nocellara del Belice cv.), was passed through a polystyrene absorbing resin column (Mitsubishi Chemical, Weekday, Japan Standard Time) which is able to absorb polyphenol compounds [41]. Then, the entrapped polyphenols were eluted by modifying their external conformation by a mild KOH solution and passed through a cationic resin to re-establish the natural pH of 2.0–3.0. Finally, the phytocomplex was vacuum dried to obtain *O. europaea* L. leaf extract concentrated in polyphenols (OLECp), with a humidity less than 4.0%.

### 2.5. Determination of Total Phenolic Content

Total phenolic content of OLECp was quantitated using the Folin–Ciocalteau colorimetric modified method in accordance with Araniti et al. [42]. An amount of 200 μg/mL of stock solution of extracts was prepared with ace/w (50/50, % *v*/*v*). 100 μL of stock extract solution was mixed with 500 μL of 2 N Folin–Ciocalteu’s phenol reagent (5 min of incubation) and 400 μL of 10.75% *w*/*v* anhydrous sodium carbonate (*w*/*v*) (incubation for 25 min). After 25 min, the absorbance against blank was measured at 760 nm in a UV-Vis spectrophotometer (Multiskan GO, Thermo Scientific, Denver, CO, USA). Standard solutions of gallic acid (0, 25, 50, 100, 200 and 300 μg/mL) were prepared instantly before the utilize and measured through the above-described procedure.

The total phenolic content of the extract was expressed as mg gallic acid equivalents (GAE)/g dry weight. The experiments, performed in triplicates, were determined using the gallic acid standard solutions (y = 0.0033x; R^2^ = 0.9991).

### 2.6. Determination of Total Flavonoid Content

The total amount of flavonoid of OLECp was quantitated spectrophotometrically. Briefly, 100 μL of a stock extract solution (200 μg/mL) was mixed with 30 μL of NaNO_2_ 5% and incubated for 5 min at RT. Subsequently, this solution was mixed up with 30 μL AlCl3 10% and incubated for 5 min at RT. Subsequently, 200 μL of NaOH (1M) were added to the solution and incubate for 5 min at RT. Finally, the solution was vortexed and incubated for 10 min at RT. After the incubation, the absorbance against blank was measured at 513 nm in a UV-Vis spectrophotometer (Multiskan GO, Thermo Scientific, Denver, CO, USA). Rutin at different concentrations (0, 20, 40, 60, 80, 100, 200 and 300 μg/mL) (y = 0.0016x; R^2^ = 0.99847) was used to calculate the standard curve. The total flavonoid contents in plant extracts were calculated as mg Rutin equivalents (RE)/g dry weight. The experiments have been carried out in triplicates.

### 2.7. DPPH Assay: Radical-Scavenging Activity

Free radical scavenging potential was investigated by a modified 1,1-diphenyl-2-picrylhydrazyl (DPPH) assay, a useful method commonly used to predict the scavenging activity of an extract. Briefly, 10 μL of six different concentrations of OLECp and standard (0, 0.25, 0.5, 1.0, 2.0 and 5.0 mg/mL) were mixed with 950 μL of methanolic DPPH solution (40 mg/μL). Following 30 min at 25 °C in the darkness, the absorbance was measured at 517 nm in a UV-Vis spectrophotometer (Multiskan GO, Thermo Scientific, Denver, CO, USA) versus reagent blank. Ascorbic acid was used as positive control. Results were expressed as inhibition % and IC_50_ value, representing the concentration of the extract necessary to scavenge the 50% of DPPH radicals. The assay was performed in triplicate.

### 2.8. Evaluation of Radicals Scavenging by Electron Paramagnetic Spectroscopy (EPR)

The antioxidant activity of OLECp was also evaluated using Electron Paramagnetic Spectroscopy (EPR), a technique which allows to detect, identify and measuring radical species.

Firstly, the DPPH radical scavenging ability of OLECp was assessed. To 200 μL of methanolic DPPH solution (0.1 mM), 50 μL of the extract (5 mg/mLconcentrations each) was added, mixed, then EPR was performed after 1 min of reaction. Ascorbic acid was used as a positive control. Spectra were acquired using a Bruker Magnettech ESR5000 (Bruker Biospin MRI GmbH, Ettlingen, Germany), using the following experimental parameters: 9.43 GHz X-band, 0.05 mT modulation amplitude, 336.64 mT central field, 12.00 mT sweep, 30 s sweep time, modulation frequency of 100 Khz, 20 mW microwave power and 8 accumulations.

Moreover, the ability of the OLECp to reduce hydroxyl radical (OH^•^) was determined. Due to his very short half-life (10^−9^ s), a technique able to “trap” the free reactive oxygen species (ROS), known as spin trapping, has been developed. In this technique a nitrone or a nitrous compound can react with a free radical to produce a nitroxide more stable than free radical. The spin trap 5-tert-butoxycarbonyl-5-methyl-1-pyrroline-N-oxide (BMPO) was used to generate adducts with ROS species which are spectroscopically detectable, and also stable. The hydroxyl free radical was produced from Fenton reaction, that allow the free radical formation without the use of Fe^2+^ as a catalyst. A typical measure of hydroxyl radical from Fenton reaction consists in mixing 15 μL of the BMPO solution (1.5 mg have been dissolved in 5 mL of ddH_2_O), 75 μL of 1 mM H_2_O_2_, 75 μL of 100 μM Iron(II) sulfate heptahydrate (FeSO_4_ • 7H_2_O) and 50 μL of ddH2O. EPR acquisition were performed 1 min after mixing. OLECp was dissolved in absolute ethanol to a concentration of 5mg/mL and then (75 μL) added to reaction mixture after the hydroxyl free radical generation. Ascorbic acid (5mg/mL) was used as positive control. BMPO (B568-10) was purchased from Dojindo EU GmbH (Munich, Germany), while FeSO_4_ • 7H_2_O (7782-63-0) was purchased from Merck KGaA (Darmstadt, Germany). Hydroxyl Radicals EPR assay was performed with the same parameters above-mentioned, except for the microwave power (6 mW).

The integration of EPR spectral area was performed using OriginPro 2018 (OriginLab Corporation, Northampton, MA, USA) to evaluate the concentration of free radicals in each acquisition, as described by Rebelo et al. [43].

### 2.9. Cell Culture

Rat Morris hepatoma derived cell line (McA-RH7777) was obtained from the American Type Culture Collection (ATCC), Rockville, MD~CRL 1601. Cells were grown in Dulbecco’s modified Eagle’s high glucose medium (Dmem W/Glutamax-I, Pyr,4.5g Glu-31966047-Gibco) supplemented with 10% *v*/*v* fetal bovine serum (Fbs, Qualified, Hi,10500064-Gibco), 10,000 U/mL penicillin and 10 mg/mL streptomycin (Penicillin Streptomycin Sol, 15140122-Gibco) at 37 °C under 5% of CO_2_, in a humidified 95% atmosphere.

### 2.10. Free Fatty Acid (FFA) Exposure and O. europaea Leaf Extract Treatment

Cells in the experimental groups were grown in Dulbecco’s modified Eagle’s high glucose medium (DMEM), 1% FBS supplementation, 10,000 U/mL penicillin and 10 mg/mL streptomycin (Penicillin Streptomycin Sol, 15140122-Gibco) at 37 °C under 5% of CO_2_, in a humidified 95% atmosphere. Oleic acid (O1008-Sigma-Aldrich, St. Louis, MI, USA) 240 mM stock solution was prepared by dissolving FFA in 100% Ethanol. The cells were exposed for 24 h to 100 μM concentration of exogenous FFA. FFA has been complexed with bovine serum albumin (BSA) 33.3 μM at a 3:1 molar ratio, because of albumin concentration is a powerful factor to determine the concentration of FFA available. OLECp was dissolved in a mixture ace/w (50/50, % *v*/*v*), to a stock concentration of 80 mg/mL The cells exposed to oleic acid were treated with increasing concentrations (25 μg/mL, 50 μg/mL, 100 μg/mL and 150 μg/mL) of OLECp.

### 2.11. MTT Assay

Cell viability was assessed using the MTT colorimetric assay. When taken up by living cells, MTT is converted from a yellow to a water insoluble blue-colored precipitate by cellular dehydrogenases [44]. Briefly, 1 × 10^4^ cells/well were plated in 96 well dish and allowed to adhere over-night. The following day cells were mioand different concentration (25 μg/mL, 50 μg/mL, 100 μg/mL and 150 μg/mL) of the extract for 24 h. After 24 h of treatment, 0.5 mg/mL of MTT was added to the medium with subsequent incubation at 37 °C for 4 h. The resulting blue formazan crystals were solved in DMSO. The absorbance of each well was read on a microplate reader (Multiskan GO, Thermo Scientific, Denver, CO, USA) at 570 nm and 690 nm (Blank). The absorbance of the untreated controls was taken as 100% survival.

### 2.12. Fluorometric Determination of Intracellular Fat Content

To evaluate the intracellular triglycerides accumulation, Nile Red staining was used [45]. Briefly, 1 × 10^4^ cells/well were plated in a 96-well dish and allowed to adhere over-night. The following day cells were exposed to 100 μM of oleic acid and different concentration (25 μg/mL, 50 μg/mL, 100 μg/mL and 150 μg/mL) of the extract for 24 h. After 24 h of treatment, medium has been removed and the cells cells has been washed with PBS. Then, the cells were incubated with 0.75 μg/mL AdipoRedTM Reagent (PT-7009, Lonza, Basel, Switzerland) dye for 15 min at RT. Fluoroskan Ascent Microplate Fluorometer (Thermo Fisher Scientific, Waltham, MA, USA) was used to determine Nile red intracellular fluorescence (485 nm excitation and a 535 nm emission).

### 2.13. Determination of Intracellular Total Fatty Acid Content

Neutral lipids, stored into lipid droplets (LDs), were visualized by Oil Red O cell staining (ORO 1.02419-Sigma-Aldrich, St. Louis, MI, USA) [45]. Briefly, 5 × 10^4^ cells/well were seeded in a coverslip and treated with FFA and different concentration (25 μg/mL, 50 μg/mL, 100 μg/mL, 150 μg/mL) of the extract for 24 h. Following two PBS washing, Cells were fixed with 3% paraformaldehyde for 15 min. Intracellular lipids were stained with Oil Red O Stain (3.3 μg/mL) for 8 min. Cell nuclei were stained with Mayer’s Hematoxylin dye (Mayer’s Hematoxylin-05-06002/L-Bioptica, Milan, Italy) for 4 min or incubated with DAPI (D8417, Sigma Aldrich, Milan, Italy). The staining procedures were performed at RT.

### 2.14. Fluorescence Image Acquisition

Fluorescence image acquisitions were carried out through a confocal microscope TCS SP5 (Leica Microsystems, Wetzlar, Germany). ORO fluorescent images were acquired with a 63X objective and a minimal number of five images for each group were taken and processed.

ImageJ Fiji (version 2.3.0/1.53f, WS Rasband, National Institute of Health, Bethesda, MD) was used to perform lipid droplets analysis. The minimum threshold value (90–255) was selected and kept constant to quantify the positive pixels percentages which were used as lipid droplets quantification.

### 2.15. Cytokines Bioplex Assay

An amount of 1.5 × 10^6^/plate McA-RH7777 cells was seeded in a 100 mm cell culture dishes and treated as previously described. Cells were washed with ice-PBS and protein were extracted through a RIPA lysis buffer (50 mM Tris-HCl, pH 8, 0.15 M NaCl, 0.5% sodium deoxycholate, 1% Triton X-100, 0.1% sodium dodecyl sulfate). Following a centrifugation at 16,000× *g* for 20 min, the supernatants were collected and stored until testing. Total protein concentration has been assessed by Bradford assay (Bio-Rad #50 Hercules, CA, USA). Protein lysate was used for quantitate simultaneously Tumour Necrosis Factor-alpha (TNF-α), eotaxin, multiple interleukins (IL-1α, IL-1β, IL-2, IL-3, IL-4, IL-5, IL-6, IL-9, IL-10, IL-12 p40, IL-12 p70, IL-13 and IL-17), interferon (IFN-ɣ), granulocyte colony stimulating factor (G-CSF), granulocyte-macrophage colony stimulating factor (GM-CSF), keratinocyte chemoattractant (KC), macrophage inflammatory protein (MIP)-1α, MIP-1β, regulated upon activation normal T cell expressed and secreted (RANTES), monocyte chemoattractant protein 1 (MCP-1), through a Bio-Plex cytokine assay (Bio-Rad, Hercules, California, USA), according to the manufacture’s protocol. The Bio-Plex suspension array system was used to read the beads and all data were analysed using Bio-Plex Manager^TM^ software (Bio-Rad, Hercules, CA, USA) [46].

### 2.16. Statistical Analysis

Data were analysed using GraphPad PRISM 9.3.1 (GraphPad Software, Inc., La Jolla, CA, USA). The results are shown as mean ± S.E.M. Shapiro–Wilk test was using to test normality. Data with normally distribution were analysed by one way ANOVA followed by Tukey’s test, whereas data not normally distributed were analysed using Kruskal–Wallis followed by Dunn’s tests. Comparisons of data derived from two groups were performed with the Unpaired Two-tailed Student’s t test or Mann–Whitney test. *p*-Values < 0.05 were considered statistically significant. DPPH assay data were fitted using nonlinear regression to compute the IC_50_ values.

## 3. Results

### 3.1. Identification and Quantification of Oleuropein and Luteolin-7-O-Glucoside Content by HPLC

HPLC analysis showed the presence of the secoiridoid oleuropein and the flavonoid luteolin-7-O-glucoside in both leaf extracts from the Nocellara del Belice and Carolea cultivars. The leaf acetone-water extract from Nocellara cv. showed a higher content of oleuropein (49.838 mg/g of leaves weight, Figure 2) and luteolin-7-O-glucoside (4 mg/g of leaves weight, Figure 2) compared to the leaf acetone-water extract of Carolea cv. that highlighted an oleuropein content of 10.827 mg/g (Figure 2) and a luteolin-7-O-glucoside content of 1.36 mg/g (Figure 2).

The leaf hydroalcoholic extract from Nocellara del Belice cv. showed an oleuropein content of 39.867 mg and 3.79 mg of luteolin-7-O-glucoside compared to the hydroalcoholic leaf extract of Carolea, that highlighted an oleuropein content of 9.55 mg/g (Figure 3) and a luteolin-7-O-glucoside content of 1.29 mg/g of leaves weight (Figure 3).

### 3.2. Analysis of Olea europaea L. Acetone-Water Leaf Extract Concentrated in Polyphenols

*O. europaea* L. leaf extract concentrated in polyphenols (OLECp), was characterized qualitatively and quantitatively by HPLC analysis for the presence of oleuropein and luteolin-7-O-glucoside. The concentrated extract revealed that 1 g of extract contained 315.250 mg of oleuropein (Figure 4) and 17.44 mg of luteolin-7-O-glucoside (Figure 4).

### 3.3. Characterization of Total Phenolic and Flavonoid Content of Olea europaea L. Leaf Extract Concentrated in Polyphenols

Phytochemical screening showed that the polyphenols concentrated leaf extract had an amount of phenolic compounds equal to 92.93 ± 9.35 mg GAE/g of extract, whereas the flavonoid content was of 728.12 ± 16.04 mg RE/g.

### 3.4. In Vitro Antioxidant Activity of Olea europaea L. Leaf Extract Concentrated in Polyphenols

The antioxidant capacity of the extract was firstly evaluated by the DPPH assay. A nonlinear regression has been used to explain the strong relation between concentration and percentage inhibition was explained by. As regards OLECp (IC_50_ = 2.30 ± 0.18 mg/mL, Figure 5) exerted a radical scavenging potency similar to ascorbic acid (IC_50_ = 1.85 ± 0.22 mg/mL, Figure 5).

### 3.5. Evaluation of Radicals Scavenging Activity by EPR Spectroscopy

Furthermore, radicals scavenging activity of OLECp was assessed by DPPH-EPR test and EPR spin trapping technique for hydroxyl radical.

For DPPH-EPR test, a six-line pattern DPPH spectrum was obtained, with an integrated spectral area value of 264.79 a.u. (Figure 6). OLECp (5 mg/mL) reduced the DPPH signal intensity (∫ = 22.51 a.u., Figure 6). Ascorbic acid (5 mg/mL), due to its high antioxidant power, was used as positive control (∫ = 32.33 a.u., Figure 6).

Concerning hydroxyl scavenging activity evaluation, EPR spectrum of the BMPO-OH adduct highlighted the typical 4-line absorption and the integration of spectral area showed a value of 47.637 a.u. (Figure 7A). The addition of the OLECp (5 mg/mL) neutralized the hydroxyl radical concentration generated through Fenton reaction, as confirmed by the decreased spectral area (∫ = 14.628 a.u., Figure 7B). Ascorbic acid (5 mg/mL) was used as positive control (∫ = 16.17 a.u., Figure 7C).

### 3.6. Olea europaea L. Leaf Extract Treatment Does Not Affect the Cell Viability

To examine whether the exposure to oleic acid and treatment with OLECp or vehicle (ace/w, 50/50, % *v*/*v*) impair the metabolic activity in McA-RH7777, an MTT assay was performed. The reduction of MTT into insoluble formazan occurs throughout a cell and it could be affected by several factors, such as metabolic and energy perturbations. The exposure to 100 μM of Oleic Acid (OA) and increasing concentrations (25 μg/mL, 50 μg/mL, 100 μg/mL, 150 μg/mL) of OLECp did not affect the hepatoma cells’ viability (Figure 8). Moreover, cells were treated with the same amount of acetone used for each extract concentration, to evaluate the effect of the vehicle in which the extract has been solubilized. The cell metabolic activity was significantly reduced after 24-h of incubation with a % acetone equivalent to 0.16 % (*p* < 0.05 vs. 100 μM OA Figure 8) and 0.24% (*p* < 0.05 vs. 100 μM OA Figure 8); however, the vehicle equivalent concentration of 100 μg/mL and 150 μg/mL, respectively, was able to produce a significant reduction in cell metabolic activity respect to the cells treated with OA 100 μM.

### 3.7. Olea europaea Leaf Extract Concentrated in Polyphenols Reduces Intracellular Lipid Accumulation

McA-RH7777 exposed to 100 μM of OA for 24 h had a significant cytoplasmic lipid droplets accumulation, highlighted by an intense red coloration under visible light or strong red fluorescence (*p* < 0.001 vs. CTRL, Figure 9), that was significantly reduced by OLECp at the concentrations of 50 and 100 μg/mL, (*p* < 0.05 vs. OA, Figure 9) as shown by Oil Red O staining. In addition, Nile Red assay underlined a reduction in lipid accumulation also at 25 μg/mL (*p* < 0.05 vs. OA, Figure 10).

### 3.8. Olea europaea Leaf Extract Concentrated in Polyphenols Improves Inflammation Status Induced by Exposure to Oleic Acid

The Bio-Plex assays allows to quantitate the level of multiple cytokines in cultured cells. An entire cytokines panel has been analysed in cellular protein extract after 24 h stimulation with OA 100 μM and co-treatment with OLECp at different concentration. McA-RH7777 exposed to OA 100 μM revealed a general inflammatory status which was attenuated by the treatment with OLECp (Figure 11).

## 4. Discussion

Leaves from *O. europaea* L. are an abundant, renewable and less-expensive source of phenolic compounds which could be affected by several factors, such as geographical origin, harvesting period and cultivar, and by technological parameters employed for the extraction, such as technique, chemical properties of the compounds and used solvent [47].

In our study, a 3-h maceration protocol followed by sonication, to maximize the total phenolic content from *Olea europaea* L. leaves, has been performed [33]. Ultrasounds ensures a higher extractive value, compared to traditional extraction method, such as Soxhlet extraction or conventional maceration [48].

Indeed, several studies have demonstrated that sonication improves the extraction performance, using less solvent, lower and controlled temperatures and shorter extraction time, positively affecting the extraction of thermolabile and unstable compounds with an increased extraction yield [49,50].

The polarity of the solvent has an important role in the selective extraction of phenolic compounds, with ethanol, methanol, acetone, as the most used solvents for their extraction; several studies showed that mixtures of organic solvents can lead to higher oleuropein recoveries compared to pure solvents [33,51]. In our study, oleuropein levels was affected by different solvents in the same harvest-time and cultivar. Indeed, HPLC underlined that the acetone/water (50/50, % *v*/*v*) extract had the major content in oleuropein, although other authors described aqueous methanol and ethanol among the most favourable solvents for extracting oleuropein from olive leaves [52]. However, it was shown that using acetone/water (50/50, % *v*/*v*) mixture during sonication allows to obtain a total phenolic content significantly higher compared to the mixtures of ethanol/water (50/50, % *v*/*v*) or methanol/water (50/50, % *v*/*v*) [33]. Oleuropein in *O. europaea* leaves has been widely described and its content depend on plant genotypes. In our study, Nocellara del Belice cv. showed the highest content of oleuropein with respect to the Carolea cv., and within the range of oleuropein content observed in other cultivars. Di Meo et al., in line with our data, showed that oleuropein content varies considerably with cultivars and harvesting periods. In particular, the authors showed that in Coratina cv. the oleuropein level, in the leaves harvested in June, is comparable with the levels observed in our study, although the solvents used for the extraction were different [35]. In addition, the hydroalcoholic (50%, *v*/*v*) extract from Nocellara contained oleuropein within the literature-reported range [53,54]. Unexpectedly, the lowest content of oleuropein was determined in Carolea cv., with the extract obtained using EtOH-water, although Nicolì et al. [55], reported higher level of the secoiridoid in the same cultivar. These differences could be ascribed to the different harvested period or, maturation degree of the leaf, as well as the cultivation area [56,57]. The screening of phytochemical content showed that the presence of polyphenols and flavonoids was observed in our two tested cultivars. Flavonoids represent major phenolic compounds in *O. europaea* leaves [32]. These compounds have remarkable biological properties. Among them, luteolin-7-O glucoside, the glycoside form of luteolin, a metabolite belonging to the group of flavones, was demonstrated to enhance the resolution of inflammation [58], to regulate liver lipid metabolism and to control diet-dependent metabolic diseases [59]. The production of flavonoids is predominately driven by genetic and environmental factors [60] and the results of our study also demonstrated a difference among the investigated cultivars. Indeed, between the two analysed varieties, Nocellara del Belice cv. showed a higher content in luteolin-7-O-glucoside, while leaves collected from Carolea cv. gave the lowest amount of this flavonoid. Both the acetone-water and the hydroalcoholic extracts showed a content of luteolin-7-O-glucoside in the range of some Spanish, Croatians cultivars growing in Morocco, as well as for a Nocellara del Belice cv. growing in China [53,54,61,62] and in line with Benincasa et al. [63], who investigated the leaf extract of the most common variety of Calabrian cultivars. Based on the higher level of phytochemical compounds in acetone-water leaf extract from Nocellara del Belice cv., a dry extract, rich in phenolic compounds, has been obtained, using a polystyrene resin. Strategies to recover and concentrate polyphenols could be favourable for several aspects. Among them, the purification of highly polyphenols concentrated extracts, from waste as leaves, offers novel ingredients which could be used in pharmaceutical/nutraceutical sector for disease prevention, in cosmesis and to prepare functional feeds for animals [64,65,66]. Furthermore, but to a not negligible extent, the waste from olive production represents a serious environmental issue [67]. Indeed, a negative effect has been extensively demonstrated on microbial populations of the soil, aquatic ecosystems and the air because of the emissions of phenol and sulphur dioxide [68]. Compared with the contents in the raw leaves, before the polyphenolic concentration, the content of oleuropein and luteolin-7-O-glucoside, in the final purified products, were increased 6.32-fold and 4.36-fold, respectively; with an amount of phenolic compounds equal to 92.93 ± 9.35 mg gallic acid equivalents (GAE), whereas the flavonoid content was of 728.12 ± 16.04 mg RE/g. Recently, biopolymers and polymeric absorbents have been used to separate and purify oleuropein and other bioactive compounds from *O. europaea* or other plant and natural resources [69,70]. Li et al., using an LSA-21 macroporous resin, reported that both total flavonoids and oleuropein were increased 13.2-fold and 7.5-fold in the purified extract, with a high yield recovery [71]. Furthermore, and in line with our data, Liu et al., using dynamic adsorption/desorption on a column packed with BMKX–4 resin reported that oleuropein increased 4.17-fold compared to raw leaves in the final product [72]. *Olea europaea* leaf extract concentrated in polyphenols (OLECp) demonstrating its specific scavenging activity against free radical. Its antioxidant activity has been investigated through three widely recognized assays (DPPH, DPPH-EPR and hydroxyl radical scavenging) [73,74,75].

DPPH assay is a commonly used method based on a mechanism of electron transfer with proton loss, which is reduced by receiving an electron followed by proton transfer from antioxidants compounds [76].

In our work, OLECp exerted a radical scavenging potency similar to ascorbic acid, which was used as positive control due to its high antioxidant power [77].

However, DPPH assay could be affected by the used solvent, since a slight mechanism of hydrogen atom transfer may occur [78], or by undesired coloured compounds present in the samples [79].

To overtake the DPPH-Assay issues, Electron Paramagnetic Resonance (EPR) technique was performed. The EPR spectroscopy allows to evaluate the scavenging activity of a specific extract against free radicals [78], since its capability to detect, identify and measure radical species. Free radical species have one or more unpaired electron in their outer shell [80,81], then, based on this, EPR detects the transitions of unpaired electrons in an applied magnetic field.

While the classical DPPH assay is an indirect technique affected by the issues mentioned above, the DPPH-EPR test is a straightforward method which allow a direct measurement of the radical in the presence of the antioxidant [78].

In this work, the characteristic six-line pattern DPPH EPR spectrum was obtained [82] and the spectroscopy results have indicated the highest DPPH radical scavenging capacity of OLECp, confirming its high antioxidant capability.

Moreover, the hydroxyl radical scavenging activity was evaluated using the spin trapping technique. In this method, the free radical is produced from a specific reaction. Due to the very short half-life of the generated free radical, a diamagnetic molecule (spin trap, such as BMPO) is used to generate a stable and spectroscopically detectable spin adduct with a longer half-life [43]. Our BMPO-EPR spin trapping results have highlighted the neutralizing power of OLECp against hydroxyl radical, as confirmed by the decreased signal area of the BMPO-OH spin adduct, indicating a decrease in the hydroxyl radical concentration. To the best of our knowledge, it is the first time that EPR spectroscopy has been used to assess the radical scavenging activity of an *O. europaea* leaf extract concentrated in polyphenols, particularly, against hydroxyl radical, which is recognised as the most highly and powerful reactive species among the ROS, leading to serious injury to cells and tissues [83]. The antioxidant capability of *O. europaea* leaf extract is generally related to the high phenolic content. Indeed, it has been demonstrated that the phytochemical content and antioxidant activity are directly and closely related [74,84,85,86]. Previous works showed that oleuropein and luteolin-7-O-glucoside exert their scavenging activity against both hydroxyl and DPPH radicals [87,88,89].

Moreover, it is known that polyphenols, counteracting reactive oxygen species (ROS) overproduction, exert significant antioxidant activity, coupled with other protective action, such as autophagy restoration and attenuation of apoptosis [90].

Furthermore, the *O. europaea* extracts are capable of protection against the lipid metabolism perturbation, not only by their antioxidant activity but also reducing fat accumulation in the liver and the expression of the proteins involved in the hepatic inflammation [91]. NAFLD, in the early steps, is characterized by an ectopic accumulation of fat in the liver which occurs because of an imbalance between lipid influx and decreased lipid disposal [92]. The cellular stress is produced directly by lipotoxicity or through the cellular response via the activation of some pathways, including oxidative stress, that contribute to the development of chronic inflammation, inducing progressive fibrosis leading to cirrhosis. Interestingly, it has been showed that oleic acid is more steatotic than palmitic acid [93] and it is able to induce lipid accumulation leading to reactive oxygen species (ROS) generation and injury mediated by inflammatory cytokines [94] which, all together, contribute to the progression of NAFL to NASH.

In this work, the lipid accumulation induced by OA in the hepatoma cells, confirmed by Nile Red staining and Oil Red O cell staining, led to an inflammation status which increased the level of multiple cytokines, including tumor necrosis factor alpha (TNF-α), Interleukin-1 alpha (IL-1α), Interleukin-1 beta (IL-1β), monocyte chemoattractant protein 1 (MCP-1) and Granulocyte colony stimulating factor (G-CSF). TNF-α is one of the main pro-inflammatory cytokines involved in systemic inflammation and it has been significantly related to non-alcoholic steatohepatitis (NASH) and hepatic fibrosis [95], playing a key role in the progression of non-alcoholic fatty liver disease [96]. A recent metanalysis highlighted that increased IL-1β concentrations, as well as TNF-α, and were significantly associated with NAFLD [95]. These pro-inflammatory cytokines, produced by adipose tissue, are involved in the inhibitor kappa B kinase beta/nuclear factor kappa B (IKK/NF-kB) and in the c-Jun N-terminal kinase/activator protein 1 (JNK/AP1) pathways by activating intracellular kinases [97]. Furthermore, Schroeder et al. showed that the increased levels of IL-1β induced the expression of IFN-ɣ in primary culture of rat hepatocytes, further highlighting the intrahepatic cascade production of inflammatory mediators [98]. It has been observed that the monocyte chemoattractant protein 1 (MCP-1) is upregulated in high fat diet-fed C57BL/6 J mice [94,99] and that elevated concentrations of G-CSF, a cytokine that generally promotes the production and the mobilization of neutrophils, directly affect hepatocytes through the GCSFR-SOCS3-JAK-STAT3 pathway, and indirectly regulates immune cell infiltration into the liver [100]. Overall, this evidence confirms the involvement of these inflammatory mediators in NAFLD development, suggesting that OA-induced lipid accumulation can be considered a useful in vitro model for studying the reduction in the lipid accumulation in response to a natural extract, leading to increased levels of ROS and expression of TNF-α, IL-1α, IL-1β, MCP-1 and G-CSF [101].

In our experiments, the treatment with OLECp decreased the levels of these cytokines and improved the general inflammation status, as result of a reduction in intracellular lipid content in hepatocytes. The anti-inflammatory property of OLECp could be related to the amount of polyphenols in the concentrated extract. Indeed, these results are consistent with previous studies showing that polyphenols inhibited lipid accumulation in OA-treated McA-RH7777 cells, probably by increasing beta-oxidation [102]. Moreover, OA can exert the same effects on HepG2 cells [103], as well as palmitic acid [104,105].

The anti-lipidemic and anti-inflammatory activities of our OLECp, besides the other polyphenols, could be linked with the higher percentage of oleuropein and luteolin-7-O-glucoside, as also demonstrated by their anti-adipogenic effect due to the transcriptional inhibiting of PPARγ [58,106,107] and by the anti-inflammatory activity mediated by the inhibition of IL-1β and TNF-α production in an in vitro model of obesity-induced inflammation [108]. Moreover, in an in vivo study on mice fed a high-fat diet, it has been demonstrated that luteolin shows anti-obesity properties, decreases the body and epididymal fat weight and improves vascular dysfunction, by inhibiting the ROS and TNF-α action [109]. This is in also in accordance with our previous studies showing that, polyphenols improve dyslipidaemia in NAFLD [26,110,111,112,113].

Altogether, our results shed new light on the multiple properties and applications of our extracts, such as the prevention of NAFLD development and the possibility to formulate novel nutraceuticals with antioxidant properties, destined to ameliorate human health. Moreover, we have demonstrated that our extraction and purification strategies could offer new opportunities to prevent environmental impact of agriculture productions through the utilization of by-products of olive chain and the use of green chemistry techniques to obtain the leaf extracts, thus promoting good practices to mitigate climate change.

## Figures and Tables

**Figure 1 plants-12-00027-f001:**
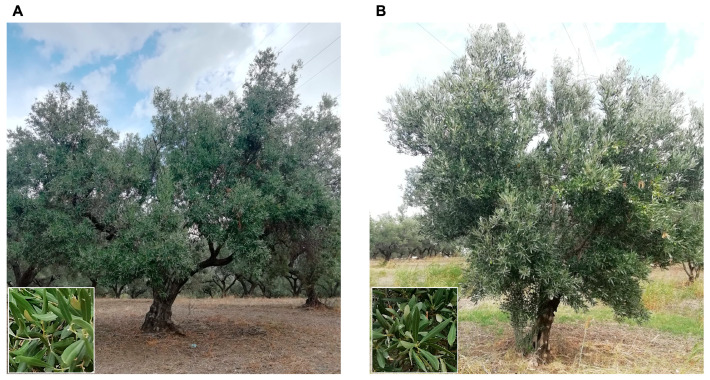
*O. europaea* L. Nocellara del Belice cultivar (**A**) and Carolea cultivar (**B**).

**Figure 2 plants-12-00027-f002:**
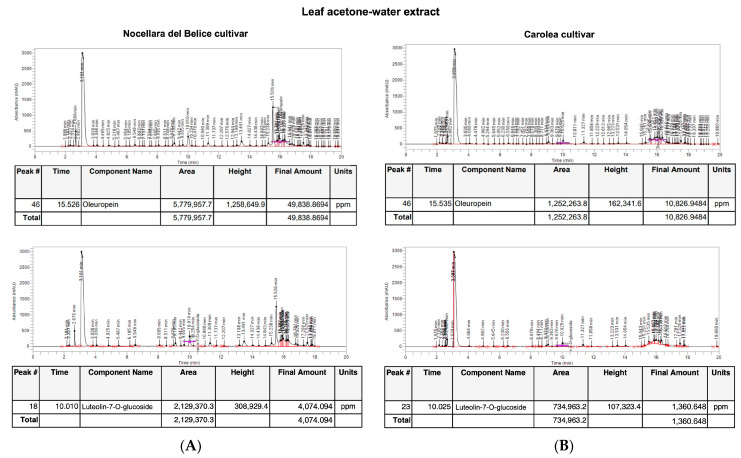
lHPLC analysis of *O. europaea* L. leaf acetone-water extract. Chromatogram of HPLC analysis of *O. europaea* L. leaf extract for (**A**) Nocellara del Belice cultivar and (**B**) Carolea cultivar.

**Figure 3 plants-12-00027-f003:**
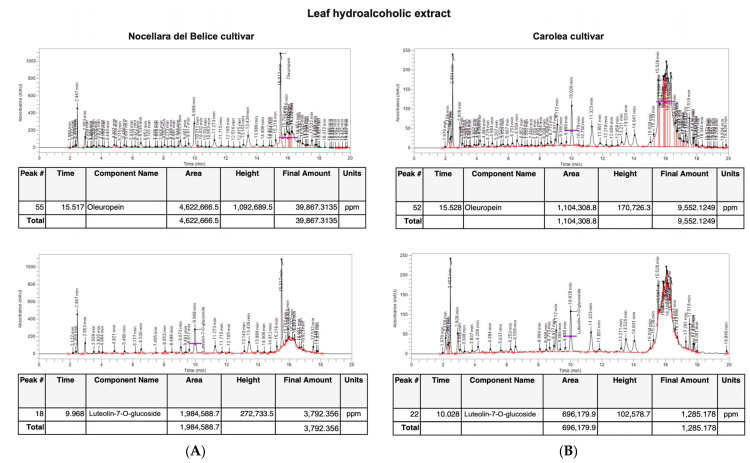
HPLC analysis of *O. europaea* L. leaf hydroalcoholic extract. Chromatogram of HPLC analysis of *O. europaea* L. leaf extract for (**A**) Nocellara del Belice cultivar and (**B**) Carolea cultivar.

**Figure 4 plants-12-00027-f004:**
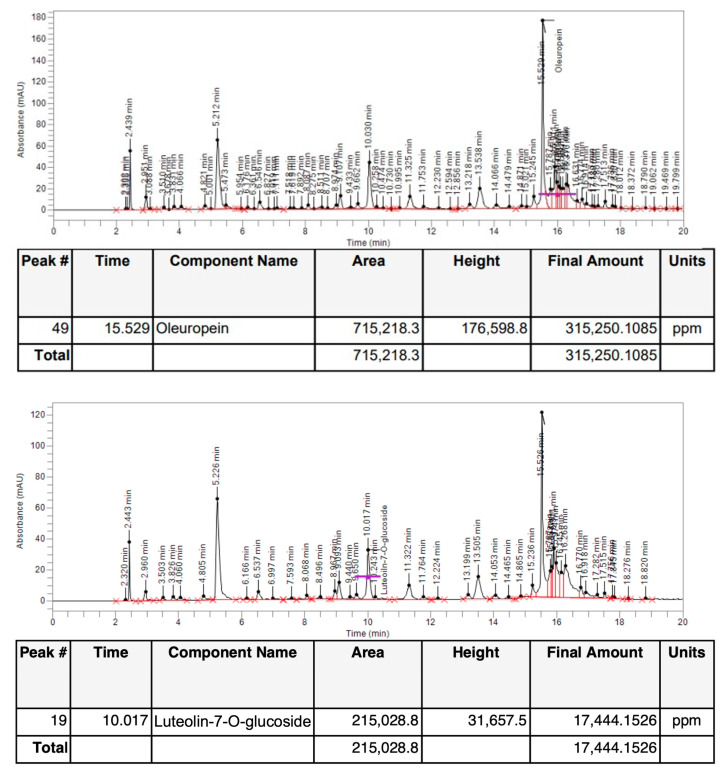
HPLC analysis of *O. europaea* L. leaf concentrated extract by resin (Nocellara del Belice cultivar).

**Figure 5 plants-12-00027-f005:**
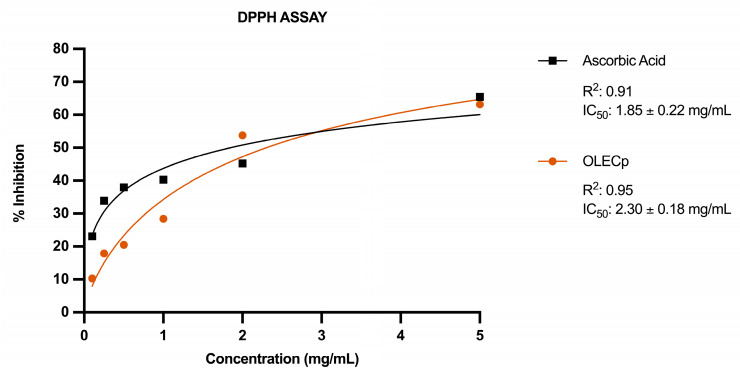
DPPH Assay. Antiradical activity (% Inhibition) and IC_50_ of OLECp and Ascorbic Acid (positive control). Data were fitted using nonlinear regression to compute the IC_50_ values. The results are expressed as mean ± SEM (*n* = 3).

**Figure 6 plants-12-00027-f006:**
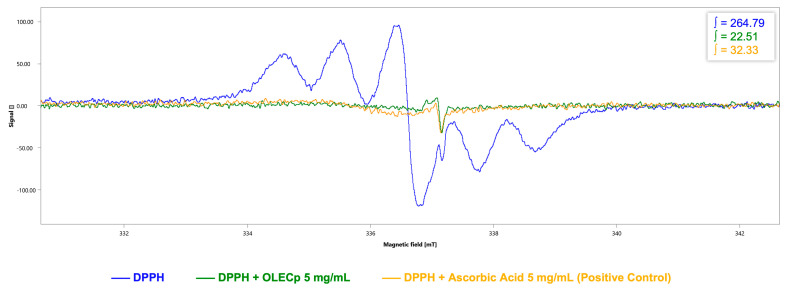
DPPH-EPR assay. EPR spectra and respective signal areas (∫) of DPPH in the absence (blue) and presence of OLECp (green) or ascorbic acid as positive control (yellow).

**Figure 7 plants-12-00027-f007:**
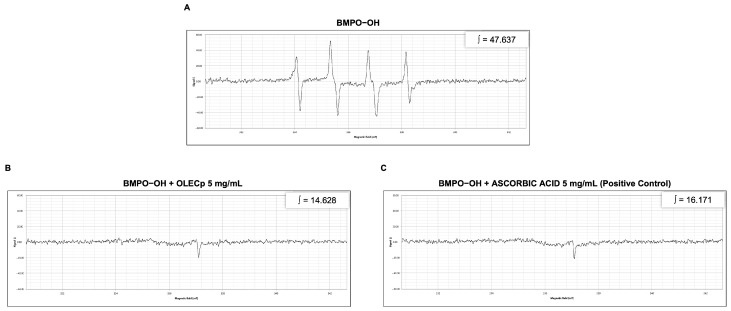
EPR spin trapping technique for hydroxyl radical scavenging activity. EPR spectra and respective signal areas (∫) of BMPO-OH adduct (**A**), BMPO-OH + OLECp (**B**) and BMPO-OH + Ascorbic Acid (**C**).

**Figure 8 plants-12-00027-f008:**
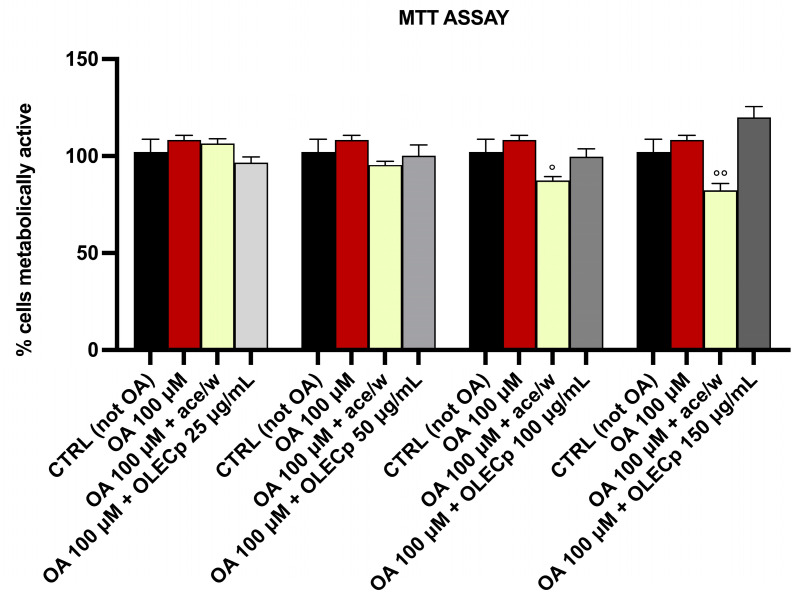
MTT Assay. Effect of 100 μM of OA, vehicle (ace/w, 50/50, % *v*/*v*) and different concentrations of OLECp after 24 h on McA-RH7777 cells metabolic activity. The results are expressed as mean ± SEM of three independent experiments; °: *p* < 0.05, °°: *p* < 0.01 vs. OA 100 μM.

**Figure 9 plants-12-00027-f009:**
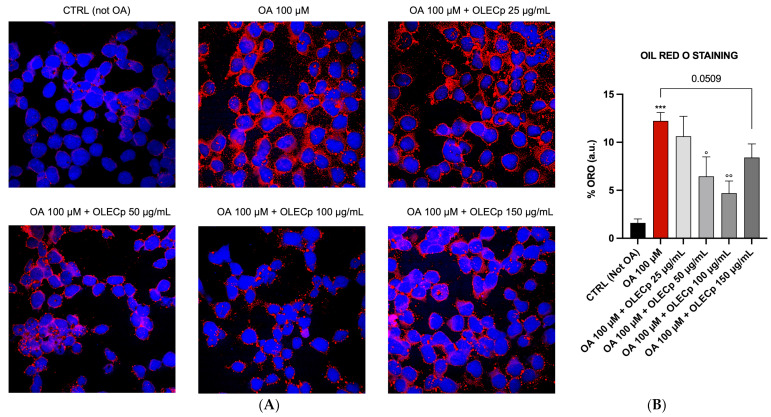
Totally fatty acid accumulation assessed by Oil Red O (ORO) Staining. (**A**) Representative ORO-DAPI merged confocal fluorescent images of intracellular lipid accumulation in McA-RH7777. (**B**) The results are expressed as mean ± S.E.M.; ***: *p* < 0.001 vs. CTRL; °: *p* < 0.05, °°: *p* < 0.01 vs. OA 100 µM.

**Figure 10 plants-12-00027-f010:**
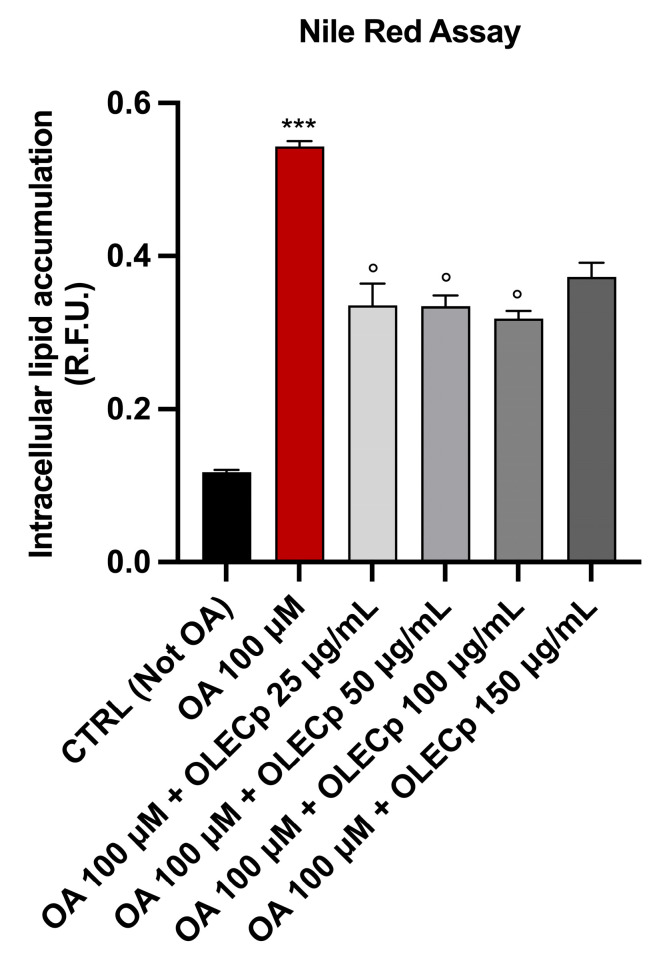
Totally fatty acid accumulation assessed by Nile Red Assay. Intracellular lipid accumulation in McA-RH7777. The results are expressed as mean ± S.E.M; ***: *p* < 0.001 vs. CTRL; **°**: *p* < 0.05 vs. OA 100 µM.

**Figure 11 plants-12-00027-f011:**
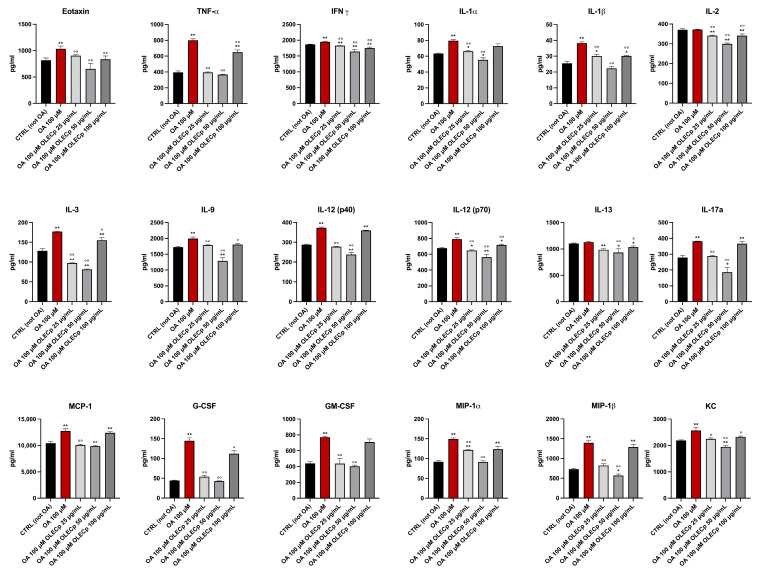
OLECp reduced pro-inflammatory cytokines expression in McA-RH7777 cells exposed to OA 100 μM. Data are expressed as mean ± S.E.M. *: *p* < 0.05, **: *p* < 0.01 vs. CTRL; °: *p* < 0.05, °°: *p* < 0.01 vs. OA 100 µM. TNF-α: tumor necrosis factor alpha, IFNγ: Interferon gamma, IL-1α: Interleukin-1 alpha, IL-1β: Interleukin-1 beta, IL-2: Interleukin-2, IL-3: Interleukin-3, IL-9: Interleukin-9, IL-12 (p40): Interleukin-12 (p40), IL-12 (p70): Interleukin-12 (p70), IL-13: Interleukin-13, IL-17: Interleukin-17a, MCP-1: monocyte chemoattractant protein 1, G-CSF: Granulocyte colony stimulating factor, GM-CSF: Granulocyte-macrophage colony-stimulating factor, MIP-1α: Macrophage inflammatory protein-1 alpha, MIP-1β: Macrophage inflammatory protein-1 beta, KC: Keratinocyte chemoattractant.

## Data Availability

The data presented in this study are available in the article. The original files are available on request from the corresponding author.

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
