# Peer review of "Nocellara Del Belice (Olea europaea L. Cultivar): Leaf Extract Concentrated in Phenolic Compounds and Its Anti-Inflammatory and Radical Scavenging Activity"

_plants, 2022, doi:10.3390/plants12010027_

Round 1
Reviewer 1 Report
In this study Olea europaea leaf extract concentrated in phenolic compounds was investigated for its anti-inflammatory and antioxidant activity.
My recommendations are as follows:
Extracts were obtained by maceration in acetone-water mixture for 3h. Therefore, the extraction procedure is conventional and conducted without optimization.
It is necessary to include an explanation of how extraction parameters were chosen. Was it based on literature?
Also, a lot of studies have been published on the optimization of extraction of olive leaves, with different techniques and solvents. Therefore, please address this in the introduction and justify why this extraction procedure was selected.
Additionally, what exactly means "sonicated in ice"? Exposed to ultrasound waves? If that is so, please include all details, such as, frequency, producer power... Additionally, why in ice?
Fig 3 contains results for hydroalcoholic extracts, however, in the extraction procedure just acetone/water extractions were mentioned.
Oleuropein and luteolin-7-O-glucoside were determined via HPLC, however, in extracts was also present a significant amount of other polyphenols, and some of them are probably known for investigated activities. Therefore, due to synergism, they probably contributed to overall activity of extract. To determine if oleuropein and luteolin-7-O-glucoside are responsible for investigated activities, it is necessary to investigate pure compounds' (oleuropein and luteolin-7-O-glucoside) activities.
As it is mentioned in the introduction, O. europaea leaves have been extensively studied and it is shown that they can be used as antioxidant and antiinflammatory agents.
And obtaining the extract of a leaf using acetone/water and concentrating after that is not an innovative method. Therefore, the introduction should be improved with a better explanation of what is actually new in this study and different from previously published ones. Also, justification for very basic extraction procedures must be included.
Author Response
In this study Olea europaea leaf extract concentrated in phenolic compounds was investigated for its anti-inflammatory and antioxidant activity.
My recommendations are as follows:
Extracts were obtained by maceration in acetone-water mixture for 3h. Therefore, the extraction procedure is conventional and conducted without optimization.
It is necessary to include an explanation of how extraction parameters were chosen. Was it based on literature?
Also, a lot of studies have been published on the optimization of extraction of olive leaves, with different techniques and solvents. Therefore, please address this in the introduction and justify why this extraction procedure was selected.
R.1 First of all, we would like to thank the reviewer for the constructive criticism and time spent to analyze deeply this manuscript. Furthermore, the observations allow us to better clarify this aspect. In our work, two different organic solvents were used for the extraction: acetone/water (ace/w, 50/50, % v/v) and ethanol/water (EtOH/w, 50/50, % v/v). Organic solvents such as methanol, ethanol/water, and acetone/water mixtures have been shown to be the best choice for extracting both lipophilic and hydrophilic phenols [1], whilst it well known that the type and the concentration of the solvent used could affect the extraction yields, highlighting an higher recovery of polyphenols as well as a major antioxidant power of the extract when 50% acetone or 50% ethanol are used [2-3].
Furthermore, our extraction consists in a 3-hour maceration protocol, using acetone/water or ethanol/water, followed by 20 minutes of sonication, to obtain a higher total phenolic content [3].
We are aware that several studies have been published on the optimization of extraction of olive leaves, with different techniques and solvents. We addressed this aspect and justified the reason of the chosen procedure, as required. The reviewer can find this part in bold red in the “introduction section”.
- Borges, A., José, H., Homem, V., & Simões, M. (2020). Comparison of Techniques and Solvents on the Antimicrobial and Antioxidant Potential of Extracts fromAcacia dealbata and Olea europaea. Antibiotics (Basel, Switzerland), 9(2), 48.
- Wang B., Qu J., Luo S., Feng S., Li T., Yuan M., Huang Y., Liao J., Yang R., Ding C. Optimization of ultrasound-assisted extraction of flavonoids from olive (Olea europaea) leaves, and evaluation of their antioxidant and anticancer activities. Molecules. 2018;23:2513.
- Irakli M., Chatzopoulou P., Ekateriniadou L. Optimization of ultrasound-assisted extraction of phenolic compounds: Oleuropein, phenolic acids, phenolic alcohols and flavonoids from olive leaves and evaluation of its antioxidant activities. Ind. Crops Prod. 2018;124:382–388.
Additionally, what exactly means "sonicated in ice"? Exposed to ultrasound waves? If that is so, please include all details, such as, frequency, producer power... Additionally, why in ice?
R.2 We thank the reviewer which allow us to better clarify this aspect. After the maceration, we performed sonication to maximize the total phenolic content from Olea europaea L. leaves. A study conducted by Mohammad B. Hossain et al. showed that the sonication allows an improvement of extraction performance [1]. Some of the advantages of the UAE include using less amount solvent, lower temperatures, and shorter extraction time, affecting positively the extraction of thermolabile and unstable compounds with an increased extraction yield. The sound waves that are produced cause the disruption of cell walls, then, the phenolic compounds are extracted [2].
We carried out sonication at controlled temperature (in ice) because sonication using a probe leads to heating of the medium because of the heat generated. This could affect the characteristics of medium, then, the cavitation. Performing sonication at lower and controlled temperature is usually necessary, indeed, keeping the temperature between 30-50° is the optimal operating condition [3].
Thus, to prevent the degradation of phenolic compounds, we have partially submerged the glass flask into a cooling bath, filled with ice [4-5]. Moreover, following the reviewer suggestion, we have now included all the details of the sonication in the “extraction procedure” section, and the reviewer can find the added part in bold red in the manuscript.
- Hossain, M. B., Brunton, N. P., Patras, A., Tiwari, B., O'Donnell, C. P., Martin-Diana, A. B., & Barry-Ryan, C. (2012). Optimization of ultrasound assisted extraction of antioxidant compounds from marjoram (Origanum majorana L.) using response surface methodology. Ultrasonics sonochemistry, 19(3), 582–590.
- Mousavi, S. A., Nateghi, L., Javanmard Dakheli, M., Ramezan, Y., & Piravi-Vanak, Z. (2022). Maceration and ultrasound-assisted methods used for extraction of phenolic compounds and antioxidant activity from Ferulago angulata. Journal of Food Processing and Preservation, 46, e16356.
- Wang B., Qu J., Luo S., Feng S., Li T., Yuan M., Huang Y., Liao J., Yang R., Ding C. Optimization of ultrasound-assisted extraction of flavonoids from olive (Olea europaea) leaves, and evaluation of their antioxidant and anticancer activities. Molecules. 2018;23:2513.
- Lavilla, I., & Bendicho, C. (2017). Fundamentals of Ultrasound-Assisted Extraction; Kingwascharapong, P., Chaijan, M. & Karnjanapratum, S. Ultrasound-assisted extraction of protein from Bombay locusts and its impact on functional and antioxidative properties. Sci Rep 11, 17320 (2021).
- Giacometti, J., Žauhar, G., & Žuvić, M. (2018). Optimization of Ultrasonic-Assisted Extraction of Major Phenolic Compounds from Olive Leaves (Olea europaea L.) Using Response Surface Methodology. Foods (Basel, Switzerland), 7(9), 149.
Fig 3 contains results for hydroalcoholic extracts, however, in the extraction procedure just acetone/water extractions were mentioned.
R.3 We thank the reviewer for pointing out the lack of the hydroalcoholic extraction in the extraction procedure section. Now we have mentioned the hydroalcoholic extraction in the manuscript and the reviewer can find this part in bold red.
Oleuropein and luteolin-7-O-glucoside were determined via HPLC, however, in extracts was also present a significant amount of other polyphenols, and some of them are probably known for investigated activities. Therefore, due to synergism, they probably contributed to overall activity of extract. To determine if oleuropein and luteolin-7-O-glucoside are responsible for investigated activities, it is necessary to investigate pure compounds' (oleuropein and luteolin-7-O-glucoside) activities.
R.4 We thank the reviewer which allow us to better clarify this aspect. The aim of our work was to evaluate the anti-inflammatory and radical scavenging activity of an O. europeae leaf extract concentrated in polyphenols (OLECp) from Nocellara del Belice. We agree with the reviewer’s assertion, indeed, the anti-inflammatory and antioxidant property of our OLECp could be related to the total amount of polyphenols in the concentrated extract. In our work, we have characterized our extract in Oleuropein and luteolin-7-O-glucoside, which represent respectively the major iridoid monoterpene and the major flavonoid in olive-leaf extract [1]. Thus, the antioxidant, anti-inflammatory and anti-lipidemic activities of our OLECp, besides the other polyphenols, could be certainly linked with the higher percentage of oleuropein and luteolin-7-O-glucoside.
Finally, we agree with the reviewer’s suggestion, and it will be interesting to carry out more extensive research relating the activities of pure compounds in the future, although that was not the intention of this work.
- [EMA (European Medicines Agency). Assessment report on Olea europaea L., folium. EMA/HMPC/359236/2016. Committee on Herbal Medicinal Products (HMPC). 2017. Available online: https://www.ema.europa.eu/en/documents/herbal-report/final-assessment-report-olea-europaea-l-folium-first-version_en.pdf].
As it is mentioned in the introduction, O. europaea leaves have been extensively studied and it is shown that they can be used as antioxidant and antiinflammatory agents.
And obtaining the extract of a leaf using acetone/water and concentrating after that is not an innovative method. Therefore, the introduction should be improved with a better explanation of what is actually new in this study and different from previously published ones. Also, justification for very basic extraction procedures must be included.
R.5 We thank the reviewer which allow us to better clarify what is new in this study and different from previously published ones. Although, it has been extensively shown that O. europaea leaf extracts can be used as antioxidant and anti-inflammatory agents, to the best of our knowledge, it is the first time that the radical scavenging activity of an O. europaea leaf extract concentrated in polyphenols (OLECp) from Nocellara del Belice cultivar has been directly assessed using Electron Paramagnetic Resonance (EPR). This spectroscopic method allows to evaluate the scavenging activity of a specific extract against free radicals since its capability to detect, identify and measure radical species. While all the classical assays are indirect techniques, EPR is a straightforward method which allows a direct measurement of the radical in the presence of the antioxidant [1]. For the first time, it has been evaluated the radical scavenging activity of an OLECp against the hydroxyl radical, which is recognised as the most highly and powerful reactive species among the ROS, leading to serious injury to cells and tissues [2].
Finally, now we have explained this in the introduction section, and the reviewer can find the added part in bold red in the manuscript.
- Giordano, A.; Morales-Tapia, P.; Moncada-Basualto, M.; Pozo-Martínez, J.; Olea-Azar, C.; Nesic, A.; Cabrera-Barjas, G. Pol-yphenolic Composition and Antioxidant Activity (ORAC, EPR and Cellular) of Different Extracts of Argylia radi-ata Vitroplants and Natural Roots. Molecules 2022, 27, 610.
- Collin F. (2019). Chemical Basis of Reactive Oxygen Species Reactivity and Involvement in Neurodegenerative Diseases. International journal of molecular sci-ences, 20(10), 2407.
Reviewer 2 Report
The manuscript ”Nocellara del Belice (Olea europaea L. cultivar): leaf extract concentrated in phenolic compounds and its anti-inflammatory and radical scavenging activity” is aimed to compare the phytochemical profile of two leaf extracts from Nocellara del Belice and Carolea cultivars. As Nocellara del Belice cv leaf extract contained the highest amount of oleuropein and luteolin-7-O-glucoside according to HPLC data, it was passed through a polystyrene absorbing resin column to concentrate polyphenolic compounds. The extract concentrated in phenolic compounds (OLECp) showed significant antioxidant and anti-inflammatory activity.
However, it is not clear why the authors used ascorbic acid as positive control to evaluate the anitradical activity of OLECp. As the authors study polyphenolics, it would be more appropriate to use quercetin.
Besides, I would recommend to rotate Figure 11 clockwise 90° so that it is more convenient for readers.
Author Response
The manuscript “Nocellara del Belice (Olea europaea L. cultivar): leaf extract concentrated in phenolic compounds and its anti-inflammatory and radical scavenging activity” is aimed to compare the phytochemical profile of two leaf extracts from Nocellara del Belice and Carolea cultivars. As Nocellara del Belice cv leaf extract contained the highest amount of oleuropein and luteolin-7-O-glucoside according to HPLC data, it was passed through a polystyrene absorbing resin column to concentrate polyphenolic compounds. The extract concentrated in phenolic compounds (OLECp) showed significant antioxidant and anti-inflammatory activity.
However, it is not clear why the authors used ascorbic acid as positive control to evaluate the anitradical activity of OLECp. As the authors study polyphenolics, it would be more appropriate to use quercetin.
R.1 We thank the reviewer which allow us to better clarify this aspect. Thanks for the suggestion to use quercetin as positive control. In our work, we used ascorbic acid, due to its recognized antioxidant activities [1]. Moreover, there are different reasons giving foundation at the rational choosing of L-ascorbic acid as standard molecule for DPPH assay.
First, the original assay was designed using this molecule. Some changes have been made over time in terms of solvents, with failed results which confirmed the better sensitivity of solutions made with methanol. In addition to this, the choice of a reference standard implies the evaluation of several parameters. Where possible, it's necessary to enlist key molecules or progenitors of the category. However, in this case, it was rationally correct to choose the molecule having the smallest carbon structure and the greatest number of hydroxyls which could represent the category without however being part of it. Therefore, in this case, it would not have been scientifically correct to use as a reference standard the same molecule whose antioxidant activity we wanted to investigate, also in consideration of the structural analogy between rutin and quercetin [2]. For this reason, ascorbic acid has been used as positive control in several work on the literature [3-4].Thus, we carried out all antioxidant assay (DPPH and EPR) keeping ascorbic acid as a positive control.
- Gęgotek, A.; Skrzydlewska, E. Antioxidative and Anti-Inflammatory Activity of Ascorbic Acid. Antioxidants 2022, 11, 1993. 775.
- Emara, K.M. (1992). Application of diphenylpicrylhydrazyl as a spectrophotometric reagent in the determination of some phenothiazines. Analytical Letters, 25, 99-109.; Sharma, O. P., & Bhat, T. K. (2009). DPPH antioxidant assay revisited. Food chemistry, 113(4), 1202-1205.
- Marrelli, M., Amodeo, V., Puntillo, D., Statti, G., & Conforti, F. (2022). In vitro antioxidant and anti-denaturation effects of Buglossoides purpurocaerulea (L.) I. M. Johnst. fruit extract. Natural product research, 1–4. Advance online publication.
- Marrelli, M., Perri, M. R., Amodeo, V., Giordano, F., Statti, G. A., Panno, M. L., & Conforti, F. (2021). Assessment of Photo-Induced Cytotoxic Activity of Cachrys sicula and Cachrys libanotis Enriched-Coumarin Extracts against Human Melanoma Cells. Plants (Basel, Switzerland), 10(1), 123.
Besides, I would recommend to rotate Figure 11 clockwise 90° so that it is more convenient for readers.
R.2 We agree with the reviewer’s suggestion. Now we have rotated the Figure 11 in the manuscript.
Reviewer 3 Report
Manuscript ID: Plants-2072950
Please check
Abstract; line 33: 92,93 and 9,35 change as 92.93 and 9.35
Keywords; line 43: please put full word: EPR; NAFLD/MAFLD; in antioxidant and anti-inflammatory delete "and"
Introduction
line 66 change 98-98,5% as 98.0-98.5%; line 139, 0.88% as 0.88%
line 68 in change waxes; and a unsaponifiable as waxes and an unsaponifiable
lines 72-75 the text is italics please change in palatine linotype
lines 117-118, 120, 151, O. europea is in palatiene change to italics check all manuscript
line 146 delete (i.e
lines 161, 174 include and in 200, 300 as 200 and 300
line 182 change 0-0.25-0.5-1-2-5 as 0, 0.25, 0.5, 1.0, 2.0, and 5.0
line 204 change tert-Butoxycarbonyl as tert-butoxycarbonyl
lines 238, 245, 255, 264, 380 include and 100, 150 as 100 and 150
References 9 and 12 please chage in journal format
Author Response
Abstract; line 33: 92,93 and 9,35 change as 92.93 and 9.35
Keywords; line 43: please put full word: EPR; NAFLD/MAFLD; in antioxidant and anti-inflammatory delete "and"
Introduction
line 66 change 98-98,5% as 98.0-98.5%; line 139, 0.88% as 0.88%
line 68 in change waxes; and a unsaponifiable as waxes and an unsaponifiable
lines 72-75 the text is italics please change in palatine linotype
lines 117-118, 120, 151, O. europea is in palatiene change to italics check all manuscript
line 146 delete (i.e
lines 161, 174 include and in 200, 300 as 200 and 300
line 182 change 0-0.25-0.5-1-2-5 as 0, 0.25, 0.5, 1.0, 2.0, and 5.0
line 204 change tert-Butoxycarbonyl as tert-butoxycarbonyl
lines 238, 245, 255, 264, 380 include and 100, 150 as 100 and 150
References 9 and 12 please change in journal format
R.1 We thank the reviewer for the suggestion. Now we have made all the requested changes, and the reviewer can find these in bold red in the manuscript.
Round 2
Reviewer 1 Report
The authors significantly improved the manuscript by including the suggested recommendations. However, there are still some suggestions, as follows:
“Hydroalcoholic solvents such as methanol, ethanol/water and acetone/water mixtures have been shown to be the best choice for extracting both lipophilic and hydrophilic phenols [31],”
In the cited work (reference 31), four extraction methods (solid-liquid, ultrasound, Soxhlet, and microwave) and six extraction solvents (water, methanol, ethanol, acetone, dichloromethane, and hexane) were used. However, the mentioned solvents mixtures were not used. Therefore, please correct this and rewrite this part by supporting your research idea.
Please avoid using the term “optimizing” and “optimization” (row 132), since extractions were not optimized, single factors were used, without determining the influence of different parameters.
Author Response
The authors significantly improved the manuscript by including the suggested recommendations. However, there are still some suggestions, as follows:
“Hydroalcoholic solvents such as methanol, ethanol/water and acetone/water mixtures have been shown to be the best choice for extracting both lipophilic and hydrophilic phenols [31],”
In the cited work (reference 31), four extraction methods (solid-liquid, ultrasound, Soxhlet, and microwave) and six extraction solvents (water, methanol, ethanol, acetone, dichloromethane, and hexane) were used. However, the mentioned solvents mixtures were not used. Therefore, please correct this and rewrite this part by supporting your research idea.
R.1 We thank the reviewer for the suggestion. We have now re-written the part and changed the reference to better support our research idea. The reviewer can find the requested change in bold red in the “introduction section”.
Please avoid using the term “optimizing” and “optimization” (row 132), since extractions were not optimized, single factors were used, without determining the influence of different parameters.
R.2 We thank the reviewer for the suggestion to avoid the term “optimizing”. We have now deleted the term “optimizing”, rearranging the sentence as follow: “The aim of our work was to compare the phytochemical profile of two leaf extracts from Nocellara del Belice and Carolea cultivars through: a) an efficient as well as less expensive extraction; b) a method to concentrate extracted polyphenols to increase the yield in bioactive compounds.” The reviewer can find the requested change in bold red in the manuscript.
Reviewer 2 Report
The manuscript can be publushed in Plants.
Author Response
We thank the reviewer.
Reviewer 3 Report
Is appropiate
Author Response
We thank the reviewer.